# The L1014F Knockdown Resistance Mutation Is Not a Strong Correlate of Phenotypic Resistance to Pyrethroids in Florida Populations of *Culex quinquefasciatus*

**DOI:** 10.3390/insects15030197

**Published:** 2024-03-15

**Authors:** Alden S. Estep, Neil D. Sanscrainte, Jason Stuck, Isik Unlu, Agne Prasauskas, Stephanie J. Mundis, Nicholas Cotter, Ana L. Romero-Weaver, Troy J. Fedirko, Natalie L. Kendziorski, Kyle J. Kosinski, Daviela Ramirez, Eva A. Buckner

**Affiliations:** 1Mosquito & Fly Research Unit, Center for Medical, Agricultural and Veterinary Entomology, United States Department of Agriculture, 1700 SW 23rd Drive, Gainesville, FL 32608, USA; neil.sanscrainte@usda.gov; 2Pinellas County Public Works—Environmental Management, 22211 U.S. Highway 19 North, Clearwater, FL 33765, USA; jstuck@pinellas.gov; 3Miami-Dade County Mosquito Control Division, 8901 NW 58th Street, Miami, FL 33178, USA; isik.unlu@miamidade.gov; 4Pasco County Mosquito Control District, 2308 Marathon Rd, Odessa, FL 33556, USA; aprasauskas@pascomosquito.org; 5Department of Geography, University of Florida, 330 Newell Dr, Gainesville, FL 32611, USA; stephanie.mundis@gmail.com; 6Center for Health Statistics, Texas Department of State Health Services, 1100 W 49th St, Austin, TX 78756, USA; 7Lee County Mosquito Control District, 15191 Homestead Road, Lehigh Acres, FL 33971, USA; ncotter@lcmcd.org; 8Florida Medical Entomology Laboratory, Department of Entomology and Nematology, Institute of Food and Agricultural Sciences, University of Florida, 200 9th St SE, Vero Beach, FL 32962, USA; aromeroweaver@ufl.edu (A.L.R.-W.); fedirko.t@ufl.edu (T.J.F.); natalie.kendziorski@gmail.com (N.L.K.); k.kosinski@irmcd.org (K.J.K.); dramirez@highlandag.com (D.R.); eva.buckner@ufl.edu (E.A.B.); 9Indian River Mosquito Control District, 5655 41st St, Vero Beach, FL 32967, USA; 10Highland Agricultural Solutions, 590 NW 3rd Street, Mulberry, FL 33860, USA

**Keywords:** *Culex quinquefasciatus*, insecticide resistance, knockdown resistance (*kdr*), CDC bottle bioassay, Florida

## Abstract

**Simple Summary:**

Previous studies have shown many populations of *Culex quinquefasciatus* are resistant to pyrethroids, the most common class of pesticide used by public health agencies. A genetic mutation has been identified that contributes to this insecticide resistance. *Culex quinquefasciatus* mosquitoes from numerous locations were tested to assess the correlation between standard resistance bioassays and this resistance mutation to determine if the mutation is a useful surrogate to assess insecticide resistance. The results from these *Culex quinquefasciatus* populations indicate that this *kdr* mutation is only a moderate strength correlate of phenotypic resistance and is thus unlikely to be a good surrogate for estimating insecticide resistance.

**Abstract:**

*Culex quinquefasciatus* is an important target for vector control because of its ability to transmit pathogens that cause disease. Most populations are resistant to pyrethroids and often to organophosphates, the two most common classes of active ingredients used by public health agencies. A knockdown resistance (*kdr*) mutation, resulting in an amino acid change from a leucine to phenylalanine in the voltage gated sodium channel, is one mechanism contributing to the pyrethroid resistant phenotype. Enzymatic resistance has also been shown to play a very important role. Recent studies have shown strong resistance in populations even when *kdr* is relatively low, which indicates that factors other than *kdr* may be larger contributors to resistance. In this study, we examined, on a statewide scale (over 70 populations), the strength of the correlation between resistance in the CDC bottle bioassay and the *kdr* genotypes and allele frequencies. Spearman correlation analysis showed only moderate (−0.51) or weak (−0.29) correlation between the *kdr* genotype and permethrin or deltamethrin resistance, respectively. The frequency of the *kdr* allele was an even weaker correlate than genotype. These results indicate that assessing *kdr* in populations of *Culex quinquefasciatus* is not a good surrogate for phenotypic resistance testing.

## 1. Introduction

*Culex quinquefasciatus* is an efficient vector of several disease agents including those causing West Nile disease, lymphatic filariasis, and Japanese encephalitis and is a worldwide target for vector control operations [1,2,3,4]. This species has posed challenges for operational control, but using the principles of integrated vector management (IVM) has been shown as the most effective way to manage mosquito populations including *Cx. quinquefasciatus* [5,6,7]. Monitoring insecticide resistance is a critical element of an effective IVM strategy as it can guide decision-making on appropriate and effective operational responses while helping to avoid interventions likely to be of low efficacy.

A 70-year history shows examples of insecticide resistance (IR) in *Cx. quinquefasciatus* to a variety of active ingredients (AIs), including larvicides and adulticides. Laboratory testing on populations from Okinawa, Japan, showed increasing resistance to DDT [8]. Studies from Peru and Ecuador around the same time showed that *Cx. quinquefasciatus* had a high level of “natural” resistance and that IR could be rapidly induced in the laboratory in as little as six generations although the specific mechanism was undetermined [9,10]. In a real-world demonstration of this same principle, Tanzanian populations taken from areas subject to intense pressure from malaria eradication by house spraying with dieldrin were 10-fold more resistant than a population collected from an untreated area [11]. The initial reports of IR in US *Cx. quinquefasciatus* populations from Texas and California were published in the 1960s [12,13]. Pyrethroids were initially effective against *Culex* populations up through the mid-1970s [14,15,16]. However, pyrethroid resistance was widely detected over the next decade, and this increasing IR was linked to preexisting DDT resistance [17,18,19,20,21]. In Florida, IR has been reported in *Cx. quinquefasciatus* populations for a few decades and appears to be widespread and frequently intense [22,23,24,25,26].

Studies of IR populations of *Cx. quinquefasciatus* have implicated both target site resistance mutations and enzymatic resistance, the two primary IR mechanisms in mosquitoes, as contributing to the observed IR phenotype (reviewed in [27]). Studies have identified SNPs that result in resistance across various insect orders by altering the binding of pesticides to the voltage-gated sodium channel or acetylcholinesterase, the molecular targets of pyrethroids and organophosphates, respectively [27,28,29]. In field populations of *Aedes aegypti*, the presence of specific knockdown resistance (*kdr*) genotypes has been shown to strongly correlate with pyrethroid resistance intensity, but this is not as clear for *Cx. quinquefasciatus* [30,31,32,33]. Two adjacent SNPs in the sodium channel result in changes of the normal leucine at position 1014 (1014L) to either a phenylalanine or rarely a serine (1014F or 1014S), and both SNPs have been found in Florida *Cx. quinquefasciatus* [24,26,33,34]. The 1014F mutation, the canonical *kdr* mutation, has been shown in laboratory studies to result in resistance to pyrethroids and DDT. An acetylcholinesterase SNP resulting in a glycine to serine substitution (119G to 119S) has been detected in *Cx. quinquefasciatus* populations in the Caribbean and shown to result in resistance to organophosphates [28,29].

Enzymatic resistance acts through the enhanced degradation of pesticides and/or enhanced transport and excretion. In one Florida population (and three others from Alabama), resistance ratios up to nearly 300-fold were described in the absence of the 1014F *kdr* mutation [25]. Mosquitoes collected from Vero Beach, Florida, in 1998 were resistant to pyrethroids, organophosphates, fipronil, imidacloprid, and spinosad, but not *Bti* [23]. This broad resistance to multiple AIs was attributed to strong enzymatic activity, and the relative importance of this mechanism seemed to be greater than the contribution from *kdr* mutations [24,25]. Studies using a variety of synergists indicated that the resistance phenotype had a large enzymatic contribution [18,26,34,35]. Our recent study of IR in Miami-Dade *Cx. quinquefasciatus,* demonstrated only a moderate correlation between phenotypic resistance and *kdr* genotypes, implicating enzymatic resistance as a large contributing factor [33].

The impetus for this study was, in part, driven by these recent IR studies in field populations of Florida *Cx. quinquefasciatus* which have shown limited phenotypic impact from this 1014F *kdr* mutation [26,33]. We wanted to conduct a larger study with samples from across the state to see if the same conclusions that were drawn from these studies in SE and SW Florida were consistent across the more than 1100 km span of Florida. We examined phenotypic insecticide resistance using the CDC bottle bioassay and *kdr* frequency in 89 *Cx. quinquefasciatus* populations from the state of Florida, including 17 populations from Miami-Dade County and 7 from Collier County, to test for any correlations [26,33]. We also conducted direct topical application on select populations from the Gulf Coast of Florida to quantify the level of resistance to permethrin observed in the bottle bioassay.

## 2. Materials and Methods

### 2.1. Mosquito Collections

Six to twelve egg rafts were collected from 89 locations across Florida (Figure 1) by local vector control personnel and research staff, and then shipped to the Florida Medical Entomology Laboratory or the Center for Medical, Agricultural, and Veterinary Entomology (CMAVE) to be reared for CDC bottle bioassay testing. Strains for topical application were similarly collected (10–50 rafts/location) by local vector control and shipped to CMAVE. Rafts were collected from open, artificial containers baited with dried plant material or, in the case of the strains colonized from Pinellas County, chicken excrement. Specific collection information is in Appendix A. Detailed collection methods, rearing procedures, and morphological identification followed the same methods as in [33]. The CMAVE laboratory *Cx. quinquefasciatus* strains were reared using a standard protocol previously described [36].

### 2.2. Phenotypic Resistance Testing

Insecticide resistance testing was conducted using the standard CDC bottle bioassay with AI-specific diagnostic doses (DDs) and diagnostic times (DTs) on F0 generation mosquitoes. This method as implemented in our laboratories has been described previously [33,37]. Briefly, four bottles were coated with technical grade permethrin at 43 µg/bottle, deltamethrin at 0.75 µg/bottle, or malathion at 400 µg/bottle, along with acetone-only negative control bottles. Mosquitoes were aspirated into bottles, and knockdown was scored at 0, 5, 10, and 15 min, and then every 15 min through 2 h as specified by the protocol. A subset of bottles (permethrin testing conducted at Florida Medical Entomology Laboratory) was monitored with a final count at 24 h to assess recovery [26]. If sufficient mosquitoes were available, all three AIs were tested. Knockdown was converted to percent mortality as per the CDC protocol. Data are found in Appendix A.

### 2.3. Topical Application

Topical application of permethrin and malathion was conducted as previously described [32,38]. Five to ten-day-old post-emergence mosquitoes from each strain were anesthetized with CO_2_, sorted on ice, and then weighed to allow an average mass per female. Females were sorted into cohorts of 10–20 and then dosed with 0.5 µL of a gravimetrically prepared permethrin concentration series using a PB600 repeater pipette with a 25 µL gas tight, blunt-tip syringe (Hamilton Company, Reno, NV, USA). The range of tested concentrations varied by strain to produce mortality between 0 and 100%. Control mosquitoes were treated with acetone only. Mortality was scored at 24 hr after application. The assay was repeated at least three times on different days. Topical bioassay data and fitting parameters are found in Appendix A. Abbott’s corrected mortality data for each strain were fitted to a 4-parameter logistic regression using PRISM v10 (GraphPad Software, San Diego, CA, USA). Median lethal doses (LD_50_), 95% confidence intervals, and fitting parameters for each strain were calculated by the software.

### 2.4. Knockdown Resistance Genotyping

The assessment of the 1014 *kdr* mutation used a previously described genotyping assay [33,39]. Individual organisms (collected from bottles after CDC bioassay testing and freeze-kill or tested as F0 organisms submitted directly by vector control personnel), averaging 43 per location (range: 24–147), were homogenized in 400 µL of deionized water and used immediately as template for a SYBR Green-based competitive PCR with variously GC-tailed primers for 1014L, 1014F, and 1014S and a common reverse primer. As this melt curve assay, like any other PCR-based assay, can be imprecise without proper controls, we included positive controls with known genotypes in each assay to ensure accurate genotyping. Positive controls were homogenate or purified DNA from the CMAVE susceptible strain (1014LL), an in-house Louisiana (LA) pyrethroid resistant strain with the 1014F mutation (1014FF), and an LA heterozygote (1014LF, confirmed by Sanger sequencing) or an artificial heterozygote created by combining a 1014LL and a 1014FF mosquito [33,39]. Each assay included a deionized water negative control.

Assays were assembled in 384-well plates on an epMotion 5750 liquid handling system (Eppendorf, Hamburg, Germany) and cycled using default “FAST” conditions on a QuantStudio 6 Flex system (Thermo Fisher, Waltham, MA, USA), followed by a 60–95 °C melt curve phase. The presence or absence of the 1014L and 1014F alleles was determined by characteristic melting temperature (Tm) peaks of 86.0 ± 0.4 °C and 82.2 ± 0.4 °C, respectively, from the positive controls [39]. The rare 1014S allele produces a Tm peak at ~84.5 °C. Heterozygosity at position 1014 was identified by the presence of a peak at both Tms. The calculation of allele frequencies was performed using the equations:$$f\left(1014L\right)=\frac{\left(2\times N\left(LL\right)\right)+\left(1\times N\left(LF\right)\right)}{\left(2\times N\left(FF+LF+LL\right)\right)}\;and\;f(1014F)=\frac{\left(2\times N\left(FF\right)\right)+(1\times N\left(LF\right))}{\left(2\times N\left(FF+LF+LL\right)\right)}$$

*Correlation analysis.* Percent mortality from the bottle bioassay at each AI-specific DT (permethrin: 30 min; deltamethrin: 60 min; and malathion: 45 min), mortality at 120 min, genotype percentages, and allele frequencies were used as input for Spearman’s correlation analysis and 95% confidence interval calculation using PRISM v10. Data and correlation information are found in Appendix A.

## 3. Results

### 3.1. Phenotypic Resistance Testing

#### 3.1.1. CDC Bottle Bioassay

Bottle bioassay testing indicated resistance to all three AIs, but the percentage of resistant populations varied. The testing of 75 populations with permethrin at the 43 µg/bottle DD and 30 min DT found that 92% were resistant (<90% down at DT) under CDC guidelines (Figure 2A and Figure 3A) [37]. Only five (6.7%)—three from Escambia County in the Florida Panhandle and two from Lee County in SW Florida—were susceptible (97–100% mortality). One population, from Palm Beach County, was susceptible but showed developing resistance (90–96% mortality). In both Lee and Palm Beach Counties, these susceptible populations represented only a fraction of the tested populations from these counties (2 of 13 and 1 of 11, respectively). We also noted varied recovery (less knockdown/mortality at 24 h versus knockdown/mortality at 2 h) in populations with extended exposure to permethrin. This recovery averaged 21% across 27 populations (Appendix A).

Testing with the type II pyrethroid deltamethrin on the same populations showed similar results (Figure 2B and Figure 3B). Only 1 population, from Shalimar in Broward County, was susceptible at the DD and DT, while 57 populations were resistant. Notably, only 7 of the 58 tested had mortality above 50% at the diagnostic time. Even at 120 min, mortality was 50% or more in only 29% (17 of 58) of the populations.

The testing of 55 populations with the organophosphate malathion at 400 µg/bottle DD and 45 min DT showed a range of susceptibilities (Figure 2C). Approximately 21.8% of populations were susceptible, 18.2% were categorized as developing resistance, and 60% were resistant. Four of the five counties with malathion susceptible populations (Broward, Palm Beach, Miami-Dade, and Lee) were along the south Florida coast (Figure 3C). Hillsborough County, home to the city of Tampa, also had a malathion-susceptible population.

#### 3.1.2. Topical Application

To confirm and quantify the resistance to permethrin that we observed in the bottle bioassay, we conducted a topical application of permethrin on six field strains from Pasco and Pinellas Counties that were provided in adequate quantity. In all six strains, we measured resistance relative to the CMAVE susceptible strain [36] (Table 1). Resistance ratios, determined by dividing the LD_50_ by the LD_50_ of the CMAVE strain, ranged from 20.3 in the Pinellas Cross Bayou population to 40.6 in the Pasco County strain. The Keller strain, collected from a wastewater treatment facility, had a RR of 34.9. The very resistant Pasco and Keller strains had LD_50_s of about 56 and 49 ng permethrin/mg mosquito, respectively. Assuming an average weight of about 2 mg, this is equivalent to a total dose of over 100 ng of permethrin needed to kill the average adult *Cx. quinquefasciatus* from these areas.

### 3.2. Knockdown Resistance Testing

More than 3300 individual *Cx. quinquefasciatus* representing 79 populations from across 18 counties (1–17 populations/county) were genotyped for *kdr* mutations at position 1014 using a melt curve assay (Figure 4A and Appendix A). We detected the 1014L and 1014F alleles but not the relatively rare 1014S allele. *Cx. quinquefasciatus* averaged 35% 1014LL, 41.9% 1014LF, and 23.1% 1014FF. Except for Escambia County, all counties had a mix of the three possible genotypes. The relative genotype percentages varied from county to county. In Escambia County, we did not detect either the 1014LF or 1014FF genotypes. At the other extreme, the 1014FF genotype was present in approximately 70% of mosquitoes from Walton and Clay Counties.

We did observe individual populations within counties with skewed genotype percentages. Five locations, three in Escambia County and one each in Lee and Broward Counties, had only the 1014LL genotype. At the other extreme, only one location in Walton County was 100% 1014FF. The 1014FF genotype was absent from 18 samples, which were the 5 locations mentioned above that were 100% 1014LL and 13 locations that were a mix of 1014LL and 1014LF.

Allele frequencies were calculated from these genotyping data for each individual sampling location (Figure 4B and Appendix A). The frequency of the 1014F allele was variable by site but was completely penetrant at only one sampling site in Walton County. The 1014L allele was absent from five sites. The statewide 1014F allele frequency was 0.44.

### 3.3. Correlation between Bottle Bioassay and kdr Genotype or Allele Frequency

The correlation analysis of the percent mortality at DT, genotype percentages, and allele frequencies indicated that there was a moderate negative correlation (−0.51) between permethrin mortality observed in the bottle bioassay and the 1014FF genotype percentage (Figure 5). The negative correlation between mortality and 1014F allele frequency was weaker (−0.34). We observed a weak negative correlation between deltamethrin-induced mortality and the 1014FF genotype or 1014F frequency (−0.25). The correlation coefficient for malathion-induced mortality was slightly above that of permethrin with the 1014FF genotype percentage (−0.60) or 1014F allele frequency (−0.48). Notably, because of the matrix comparison, we observed that mortality between permethrin and malathion or deltamethrin was moderately correlated (0.55 and 0.50).

## 4. Discussion

This study sought to thoroughly examine on a statewide scale, the correlation between IR detected at the standard CDC-specified DDs and DTs for three of the most common AIs in mosquito adulticides (permethrin, deltamethrin, and malathion), and the genetic marker of pyrethroid IR in *Culex*, the 1014 *kdr* mutation. To do this, we collaborated with numerous mosquito control programs to acquire egg rafts in sufficient quantity for CDC bottle bioassay testing from nearly 80 locations representing urban, suburban, and agricultural areas in Florida. We also conducted permethrin topical application on a few populations to quantify the IR detected using the bottle bioassay. We further conducted testing to determine *kdr* genotypes and frequencies in many of these same populations.

Our primary observations from the resistance testing portion of this study are straightforward and agree with previous studies in the SE US. Most populations of *Cx. quinquefasciatus* have resistance, often strong resistance, to pyrethroid AIs [26,27,33,35]. In this study, we found that more than 93% of the populations were resistant to permethrin and more than 98% of the populations were resistant to deltamethrin. The quantification of permethrin resistance in a subset of strains using topical application showed that this IR was intense, with resistance ratios up to 40. We also observed some level of recovery to extended exposure to permethrin as had been observed in a previous FL study, and this strongly suggests enzymatic detoxification [26,40]. The operational impact of this ability to recover from exposure is unclear and needs to be thoroughly investigated.

With respect to malathion, our testing showed that IR was much more variable than with the pyrethroids. In this study, nearly a quarter of the populations we tested were susceptible and likely to be well controlled by commercial adulticides containing OPs. About a quarter of the populations were in the CDC “developing resistance” category which may indicate that OP resistance is widely increasing. This calls for additional IR monitoring from these sites to determine if levels of IR are increasing over time.

We note one additional observation from the malathion testing; populations with susceptibility to malathion were found in Miami-Dade, Broward, Lee, Palm Beach, and Pinellas Counties. These are among the most urbanized and densely populated counties in Florida and have large mosquito control programs, yet they have some of the most susceptible populations of *Cx. quinquefasciatus*. Our previous study noted that malathion IR was more intense in industrial and agricultural areas rather than the urban areas of Miami-Dade, and this current study appears to support this [33]. In these locations, operational mosquito control is infrequent and thus it may not be the primary driver of strong OP resistance in *Culex* populations. Understanding the drivers of this IR, whether the agricultural or industrial use of OPs or other classes of chemicals that can drive enzymatic activity and cross resistance to other classes of AI, in these infrequently treated areas has clear implications for effective public health control, and defining these drivers is certainly an area for additional study [40].

The assessment of the 1014 *kdr* mutation showed, on a statewide scale, the same patterns we had previously observed locally in Miami-Dade *Cx*. *quinquefasciatus.* First, we did not observe the 1014S mutation in the testing we conducted. It appears to still be rare, as it was even during the initial detection in Jacksonville in 2009, and it is thus unlikely to be an important factor in operational control [26]. Second, we found that the frequency of the 1014F allele was not generally high across the state (0.44) and was more often found as the 1014LF heterozygote (~41%) rather than as the homozygous 1014FF (~23%). This is notably different from Florida populations of *Ae. aegypti* where the 1534C *kdr* mutation has reached near-fixation [30]. Selection for this mutation may not be as strong in *Cx. quinquefasciatus* if other mechanisms are responsible for a large portion of the IR phenotype.

The negative correlation between permethrin mortality and the 1014FF genotype was the strongest that we observed among the two pyrethroids, but it was moderate and not comparable to the strong/very strong correlation between the dilocus *kdr* genotype and permethrin LD_50_ or resistance ratio (ρ = 0.90) in *Ae. aegypti* [30,41,42]. The correlation between deltamethrin mortality and 1014FF in these *Culex* populations was even weaker. Additionally, we also observed that the correlation between the 1014FF genotype and mortality from malathion, an OP with a mode of action different than the pyrethroids, was equally as strong (−0.60) as that of permethrin and that permethrin and malathion mortality were positively correlated. Taken together, this suggests that using the *kdr* genotype is not a rigorous predictor of pyrethroid resistance intensity in these Florida *Culex quinquefasciatus*. It is clear that *kdr* plays a role and is beneficial for surviving insecticide exposure, but this data set suggests that factors other than *kdr* play a large role in the observed pyrethroid resistance, making the value of using *kdr* genotype as a surrogate or marker for strong IR in *Cx. quinquefasciatus* potentially dubious; this requires further investigation in other locations than Florida.

## Figures and Tables

**Figure 1 insects-15-00197-f001:**
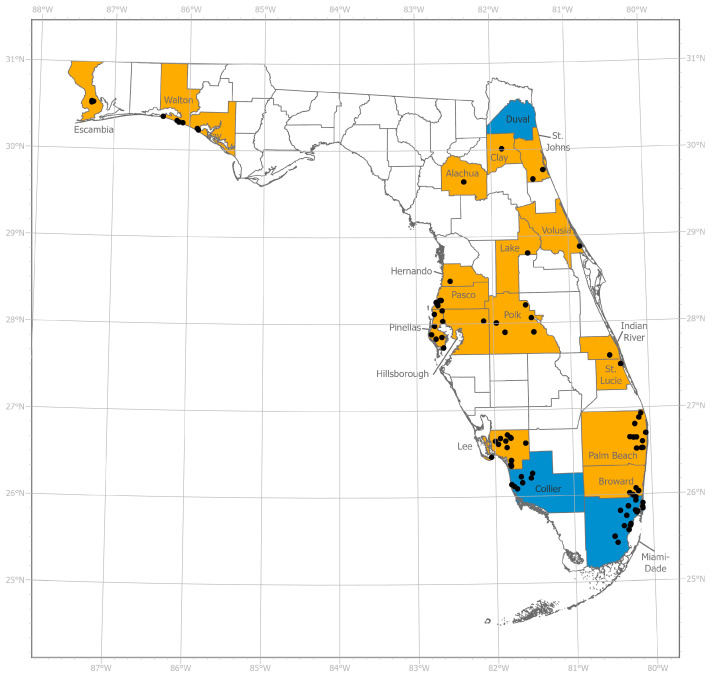
Map of sampling locations used in this and previous studies. Individual sites are marked with solid black circles and counties that provided samples are in orange. Counties highlighted in blue are locations described in previous studies of *Culex quinquefasciatus* for which knockdown resistance and phenotypic resistance testing were reported [24,26,33]. The exact location of the population tested in [24] is unclear and thus not marked with a black circle. Map created in ArcGIS Pro Version 3.1.2 using Florida county boundaries from the Florida Geographic Data Library. Detailed sample location data are in Appendix A.

**Figure 2 insects-15-00197-f002:**
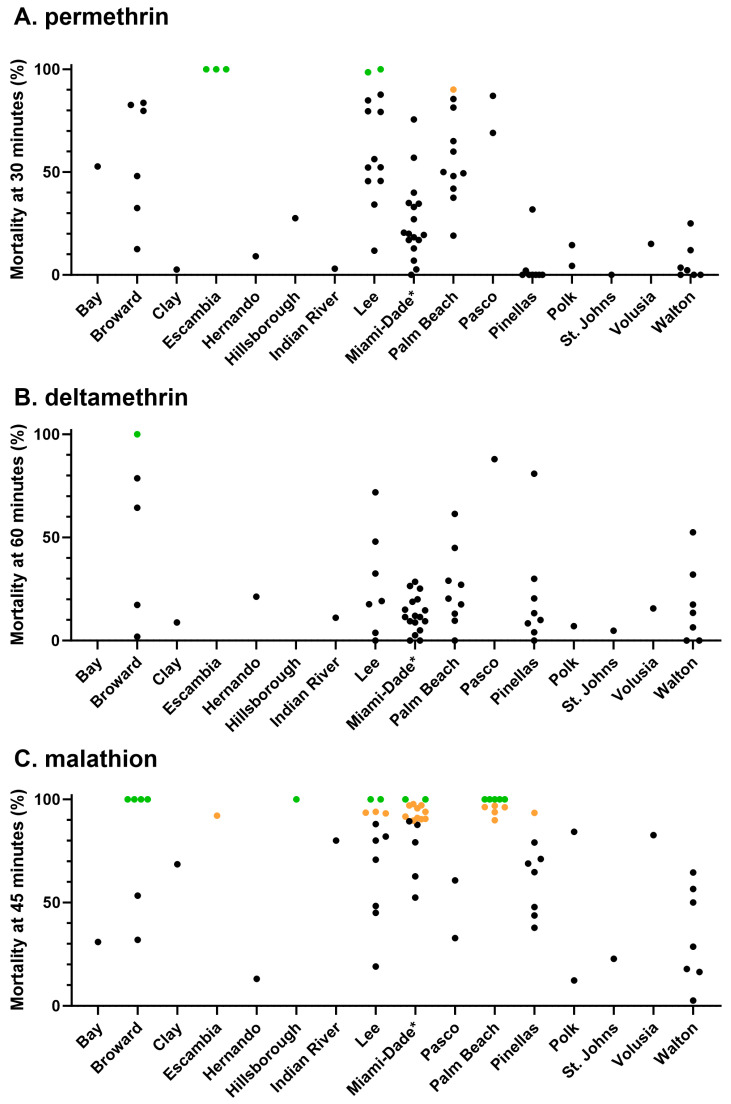
CDC bottle bioassay mortality for *Culex quinquefasciatus* collected from 16 Florida counties to (**A**) permethrin at 43 µg/bottle and 30 min; (**B**) deltamethrin at 0.75 µg/bottle and 60 min; and (**C**) malathion at 400 µg/bottle and 45 min. Results that meet the CDC definitions of susceptible, developing resistance, and resistant are colored green, orange, and black, respectively [37]. * Samples from Miami-Dade County are from [33].

**Figure 3 insects-15-00197-f003:**
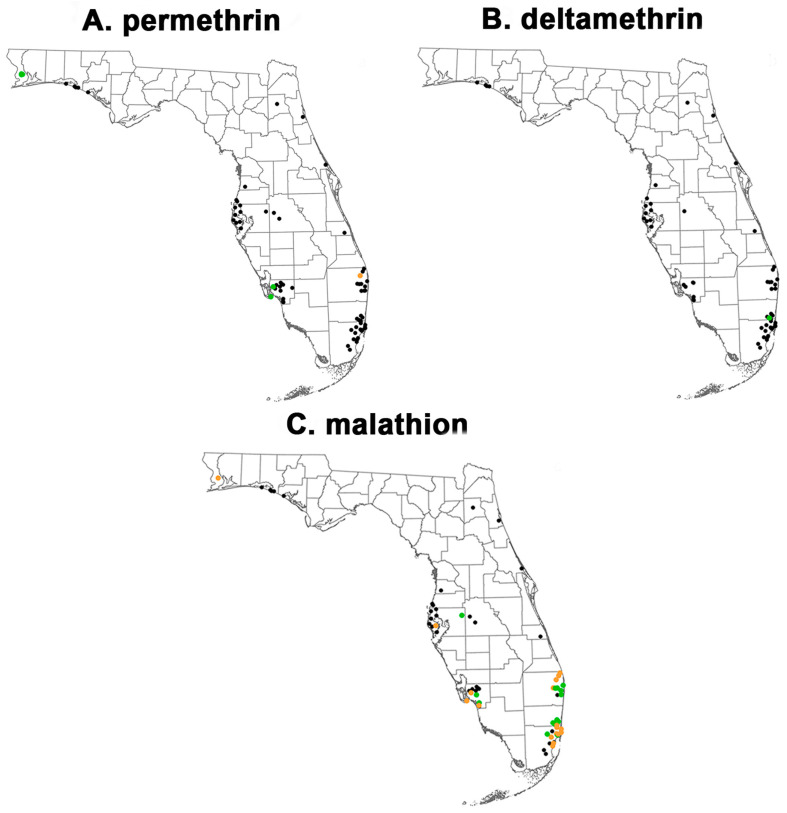
CDC bottle bioassay mortality for *Culex quinquefasciatus* collected from 16 Florida counties displayed as a map for (**A**) permethrin at 43 µg/bottle and 30 min; (**B**) deltamethrin at 0.75 µg/bottle and 60 min; and (**C**) malathion at 400 µg/bottle and 45 min. Results that meet the CDC definitions of susceptible, developing resistance, and resistant are colored green, orange, and black, respectively. Map created in ArcGIS Pro Version 3.1.2 using Florida county boundaries from the Florida Geographic Data Library. Detailed sample location data are in Appendix A. Samples from Miami-Dade County are from [33].

**Figure 4 insects-15-00197-f004:**
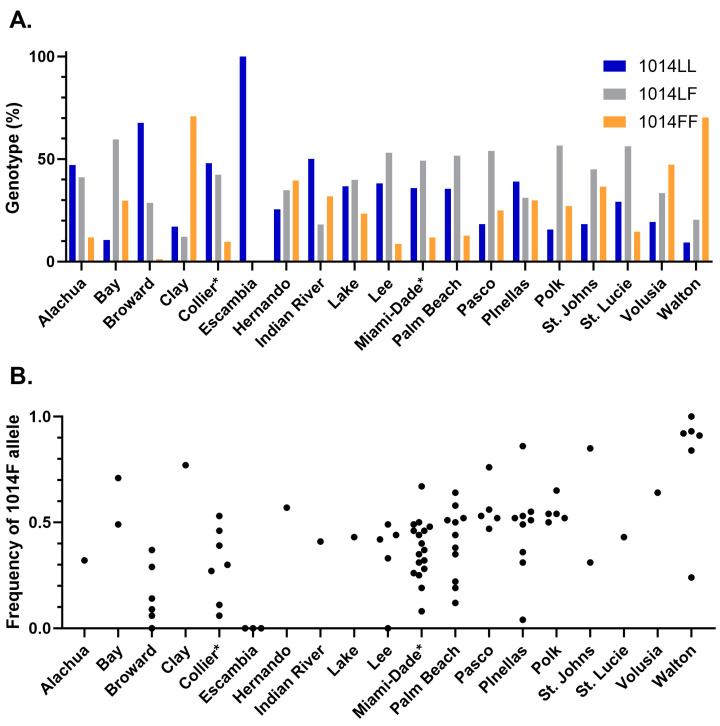
*Knockdown resistance* of *Culex quinquefasciatus* collected from 18 Florida counties showing (**A**) average genotype percentage by county; and (**B**) 1014F allele frequency where each black circle represents results from an independent collection. Genotyping was conducted using a previously described melt curve assay [33,39], and allele frequency was calculated as described in the methods. * Samples from Collier and Miami-Dade Counties are from [26,33].

**Figure 5 insects-15-00197-f005:**
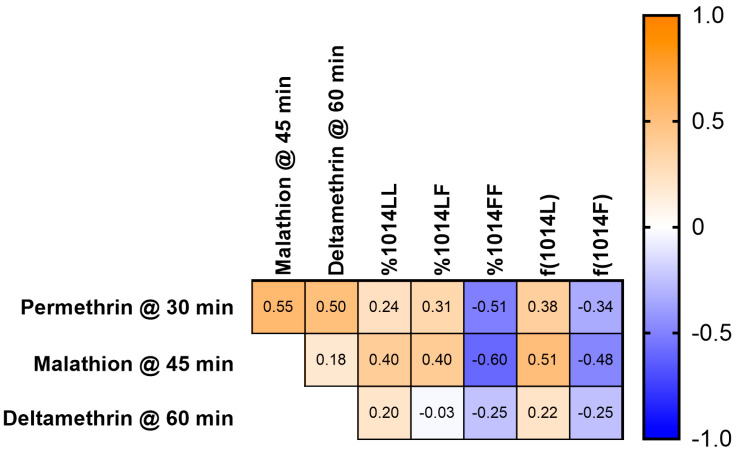
Spearman’s correlation matrix of phenotypic resistance data and *kdr* genotyping results in *Cx. quinquefasciatus* populations from Florida, USA. The matrix was constructed with data from 50 samples for which all data points were present (Appendix A), using PRISM v10, as described in the methods.

**Table 1 insects-15-00197-t001:** Median lethal dose for permethrin for the CMAVE laboratory strain and six field populations of *Culex quinquefasciatus*.

Strain ^1^	Permethrin LD_50_ ± 95% CI (ng AI/mg Mosquito)	Resistance Ratio ^2^	R^2^
CMAVE	1.39 (1.16–1.64)	1.0	0.7165
Pasco F4	56.47 (42.25–74.14)	40.6	0.9169
Cross Bayou F0	28.31 (24.06–33.75)	20.3	0.9450
North Highway F0	32.24 (21.17–50.23)	23.2	0.7067
Oldsmar Sewer F0	30.86 (22.98–42.68)	22.2	0.8291
Clearwater Nursery F0	32.98 (27.00–40.83)	23.7	0.9186
Keller F1	48.54 (39.62–58.86)	34.9	0.9223

^1^ The generation tested is noted after the strain name. F0 are adults reared from field collected rafts. F1 or F4 indicate the first and fourth generation of colonized strains. ^2^ Resistance ratio calculated by dividing the LD_50_ of the field strain by the LD_50_ of the CMAVE laboratory susceptible strain. Raw data and fitting results are in Appendix A.

## Data Availability

The data presented in this study and used in the correlation analysis are available in the manuscript, Supplementary files at https://doi.org/10.15482/USDA.ADC/25044353.v1 accessed on 6 March 2024, and/or previous publications [26,33].

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
