# Peer review of "The L1014F Knockdown Resistance Mutation Is Not a Strong Correlate of Phenotypic Resistance to Pyrethroids in Florida Populations of Culex quinquefasciatus"

_insects, 2024, doi:10.3390/insects15030197_

Round 1
Reviewer 1 Report
Comments and Suggestions for Authors
The manuscript "The L1014F knockdown resistance mutation is not a strong correlate of phenotypic resistance to pyrethroids in Florida populations of Culex quinquefasciatus" by Estep and coauthors describes field sampling of mosquitoes across Florida for testing of phenotypic insecticide resistance and evaluation of genotypes present in these mosquito populations.
Overall, the manuscript is well written, methods are appropriate, and results are clearly described. Most comments and suggestions I have are minor and indicated in the marked PDF.
The few more important comments that authors should address before publication are indicated below:
1. More detail is needed on field collections. Number (range) of egg rafts collected from field sites, procedures to ensure genetic diversity of each population for testing, lab generation tested.
2. What is the relationship between mosquitoes tested by bottle bioassay, by topical application, and by melt curve analysis? Were mosquitoes from same populations and filial generation arbitrarily selected for each procedure?
3. There is an assumption in this manuscript that the melt curve provides an accurate genotype for both homozygous and heterozygous insects. Particularly, since mosquitoes were not sequenced. What evidence is there for specificity of this genotyping method for determining kdr allele frequency in Cx. quinquefasciatus?
4. Authors should review and confirm citations provided for melt curve analysis. Provide citations for melt temperatures if these differ from citations for the melt curve methods.
5. Authors might consider moving some information from the discussion to the introduction to provide a more clear upfront rationale for the study.

Comments on the Quality of English LanguageManuscript is well written overall.
Reviewer 2 Report
Comments and Suggestions for Authors
Peer review report on the manuscript " The L1014F knockdown resistance mutation is not a strong correlate of phenotypic resistance to pyrethroids in Florida populations of Culex quinquefasciatus", (Manuscript ID: insects-2904086)
Recommendation: Accept
Comments to Authors:
This manuscript reports a study that investigates the correlation between the L1014F knockdown resistance (kdr) mutation and phenotypic resistance to pyrethroids in different populations of Culex quinquefasciatus of Florida. Despite previous associations between this mutation and insecticide resistance, the research found only moderate to weak correlations between the presence of the kdr mutation and resistance levels determined by standard resistance bioassays. This suggests that the L1014F mutation is not a reliable indicator for assessing pyrethroid resistance in these mosquito populations.
The manuscript is well-written with a well-organized text. The results are sufficiently presented and analyzed. The discussion contributes to a better understanding of the findings and highlights the complexity of insecticide resistance mechanisms, and underscores the need for comprehensive approaches to monitor and manage resistance in mosquito control programs.
Therefore, considering the following minor comments, the manuscript is recommended as it is in the present form for publication in Insects.
- Bibliographical reference No 41 (line 157 in the present text) should be renumbered No 40, and bibliographical reference No 40 (line 164 in the present text) should be renumbered No 41. This change should also be made in the references section.
- In the lines 444 and 456 the genera of Culex and Aedes should be written in italics.
Author Response
This manuscript reports a study that investigates the correlation between the L1014F knockdown resistance (kdr) mutation and phenotypic resistance to pyrethroids in different populations of Culex quinquefasciatus of Florida. Despite previous associations between this mutation and insecticide resistance, the research found only moderate to weak correlations between the presence of the kdr mutation and resistance levels determined by standard resistance bioassays. This suggests that the L1014F mutation is not a reliable indicator for assessing pyrethroid resistance in these mosquito populations.
The manuscript is well-written with a well-organized text. The results are sufficiently presented and analyzed. The discussion contributes to a better understanding of the findings and highlights the complexity of insecticide resistance mechanisms, and underscores the need for comprehensive approaches to monitor and manage resistance in mosquito control programs.
Therefore, considering the following minor comments, the manuscript is recommended as it is in the present form for publication in Insects.
- Bibliographical reference No 41 (line 157 in the present text) should be renumbered No 40, and bibliographical reference No 40 (line 164 in the present text) should be renumbered No 41. This change should also be made in the references section.
- In the lines 444 and 456 the genera of Culex and Aedes should be written in italics.
Response: Thank you for the positive comments on this manuscript. The text has been revised for both items noted above.